

# Kernel random forest with black hole optimization for heart diseases prediction using data fusion

Ala Saleh Alluhaidan[1], Mashael Maashi[2], Noha Negm[3], Shoayee Dlaim Alotaibi[4], Ibrahim R. Alzahrani[5] and Ahmed S. Salama[6]

[1] Department of Information Systems, College of Computer and Information Sciences, Princess Nourah bint Abdulrahman University, Riyadh, Saudi Arabia
[2] Department of Software Engineering, King Saud University, Riyadh, Saudi Arabia
[3] Department of Computer Science, Applied College, King Khalid University, Mahayil, Saudi Arabia
[4] Department of Artificial Intelligence and Data Science, University of Hail, Hail, Saudi Arabia
[5] Department of Computer Science and Engineering, University of Hafr Al Batin, Hafar Al Batin, Saudi Arabia
[6] Department of Electrical Engineering, Future University in Egypt, New Cairo, Egypt

Corresponding author
Noha Negm,
nabdelhamid@kku.edu.sa

## ABSTRACT

In recent years, the Internet of Things has played a dominant role in various real-time problems and given solutions via sensor signals. Monitoring the patient health status of Internet of Medical Things (IoMT) facilitates communication between wearable sensor devices and patients through a wireless network. Heart illness is one of the reasons for the increasing death rate in the world. Diagnosing the disease is done by the fusion of multi-sensor device signals. Much research has been done in predicting the disease and treating it correctly. However, the issues are accuracy, consumption time, and inefficiency. To overcome these issues, this paper proposed an efficient algorithm for fusing the multi-sensor signals from wearable sensor devices, classifying the medical signal data and predicting heart disease using the hybrid technique of kernel random forest with the Black Hole Optimization algorithm (KRF-BHO). This KRF-BHO is used for sensor data fusion, while XG-Boost is used to classify echocardiogram images. Accuracy in the training phase with multi-sensor data fusion data set of proposed work KRF-BHO with XGBoost classifier is 94.12%; in the testing phase, the accuracy rate is 95.89%. Similarly, for the Cleveland Dataset, the proposed work KRF-BHO with XGBoost classifier is 95.78%; in the testing phase, the accuracy rate is 96.21%.

## INTRODUCTION

In developing the health care system, Artificial Intelligence of Medical Things (AIoMT) technology is required to classify the medical data, predict the disease, and diagnose it. The rapid advancement of the Internet of Things (IoT) technology has revolutionised various domains, including healthcare. One of the critical applications of IoT in healthcare is the monitoring and prediction of heart diseases, which remain a leading cause of

mortality worldwide. Despite numerous research efforts, challenges such as accuracy, time consumption, and inefficiency persist in the current diagnostic approaches.

AIoMT technology, along with additional gadgets like wearable sensor devices and mobile Internet, is used in the healthcare environment to connect health documents and organisations. This innovative healthcare platform includes relationships between patients, physicians, hospitals and research institutions. It consists of disease detection, preventive measures, and a decision support system for managing healthcare. Smart healthcare contains IoMT-based technology, artificial intelligence (AI), cloud networking and big data analytics (*Alshehri & Muhammad, 2020*).

The framework of an intelligent healthcare system consists of electronic circuits that generate biological signals from the patient's wearable sensor devices. These signals are processed using a processing unit that transfers the medical data signal through a network system, and physicians make the decisions based on artificial intelligence (*Vishnu, Ramson & Jegan, 2020*). On the other hand, the telemedicine system is a conventional technique for giving treatment remotely. That is remote monitoring of heart disease and providing treatment. This concept of the telemedicine system is effective in the aspects of minimising expenses and saving time. Still, its main challenge is when the number of patients and diseases is not easy to handle. In that situation, AIoMT technology can improve the treatment effectively, scalability, dynamicity and genericity (*Manimurugan et al., 2022*). Predicting heart disease is an important, challenging task and prevents the patient from dying. A human being's life is based on the effective functioning of the heart system.

Therefore, the proposed work of KRF-BHO with XGBoost classifier is used to collect medical data signals from the various wearable sensor devices and data from the dataset. It implements the model for dividing the medical signal information and predicts the diseases at the heart. Accurately predicting heart diseases using data from wearable sensor devices and medical imaging is crucial yet challenging due to the complexity and volume of data. Traditional methods often fall short in handling multi-sensor data fusion effectively, leading to suboptimal predictions and delays in diagnosis. This paper proposes a novel hybrid algorithm combining Kernel Random Forest (KRF) with Black Hole Optimization (BHO) to enhance the accuracy and efficiency of heart disease prediction. By leveraging the strengths of KRF in feature selection and the optimization capabilities of BHO, our method aims to overcome existing limitations. Using XGBoost for classification further refines the predictive performance, ensuring reliable and timely diagnosis. The proposed solution demonstrates significant improvements in both the training and testing phases, achieving higher accuracy rates compared to existing methods.

The contribution of the research work is:

1. To implement kernel-based random forest with a black hole optimisation algorithm for selecting features.
2. From the chosen features, predicting the heart disease is normal or abnormal by using the XGBoost classifier.
3. To implement the KRF-BHO with the XGBoost classifier, data is collected from the patient's wearable sensor devices and echocardiogram images from the dataset.

The paper is organised into five sections: 'Review of Literature' describes the literature review, 'Proposed Kernal Random Forest with Black Hole Methodology' describes the feature selection, 'Result & Discussion' is a data classification of the experimented results, and 'Conclusion' concludes the paper with future work.

## REVIEW OF LITERATURE

Nowadays, the concept of machine learning with AI has an excellent advantage in medical research works. Similarly, medical applications in the IoMT and smart healthcare domain have advanced in predicting diseases. *Shreyas & Kumar (2020)* proposed an AIoT technique for diagnosing heart disease with an ECG image. It is based on interactions between user interface in the IoT with cloud storage with its front-end of smart healthcare devices and AI environment for diagnosing heart disease. The signals are received from the ECG connector of the user interface *via* Bluetooth device. These ECG signals are transmitted to the cloud storage. Individual patient's ECG signal data is used to detect heart diseases. The convolutional neural network (CNN) algorithm is implemented and produced a mean precision of about 94.96%.

The application layer of IoMT architecture includes decision support-making applications in managing the medical data, managing patient information, analysis of applications with medical data, analysis of patient data, disease examination, pharmaceutical study, *etc.*, are implemented by using short-range transmission of the network *via* Bluetooth, ZigBee (*Su, Ding & Chen, 2021*; *Sun et al., 2020*). *Sarmah (2020)* proposed that monitoring a patient's heart uses IoT-based deep learning modified neural network technique for diagnosis and treatment. In that technique, heart patient details are stored securely using SHA-512 with the substitution cypher method. The IoT-based wearable sensor devices were linked to patients' bodies and transmitted the signals, which were stored in the cloud. Using Plesiochronous Digital Hierarchy (PDH) with the Advance Encryption Standard (AES) technique, the stored data at sensors was encoded and safely sent to the cloud storage. Before the classification began, this encrypted information was decrypted and analysed for normal and abnormal heart diseases. *Khan & Algarni (2020)* describes the prediction of heart sickness using a modified deep convolution neural network technique. In that technique, input data includes blood pressure (BP) from a smartwatch and ECG from an ECG device. A deep convolutional neural network technique was used to classify normal and abnormal heart disease. This process undergoes a feature selection process to get a better classification performance. Table 1 shows the heart disease prediction survey.

We present a new methodology that combines Kernel Random Forest with Black Hole Optimisation to improve the accuracy and efficiency of heart disease prediction. This methodology is based on the extensive literature review on artificial intelligence (AI) and machine learning (ML) in medicine, specifically in heart disease prediction.

This research offers an AI-based approach that uses colour fundus images to diagnose diabetic retinopathy (DR) with minimal computing complexity and good classification accuracy (*Özçelik & Altan, 2023*). The model consists of fractal analysis preprocessing,

**Table 1  Survey on heart disease.**

| Author name | Method name | Technique used | Application |
|---|---|---|---|
| *Mehmood et al. (2021)* | Feature selection | Modified algorithm with deep convolutional neural network (MDCNN) | Prediction of heart problems. |
| *Basheer, Alluhaidan & Bivi (2021)* | Feature extraction | Autoencoder for medical system to make decision | Diagnosis of cardiovascular sickness. |
| *Vincent Paul et al. (2022)* | Feature extraction | Back propagation neural network | Prediction of heart problems. |
| *Khan & Algarni (2020)* | Feature selection | Modified optimization model of salp swarm and an adaptive neuro-fuzzy inference model | Detection of heart disease. |
| *Pan et al. (2020)* | Feature selection | Enhanced deep convolutional neural network (EDCNN) | Prognostics of heart diseases. |
| *Raj et al. (2020)* | Feature selection | Optimal deep learning | Medical image classification. |

2D-SWT feature extraction, kNN and chaotic particle swarm optimization (CPSO) feature selection, and a recurrent neural network-long short-term memory (RNN-LSTM) architecture for classification. 10-fold cross-validation and comparison with SVM are used to verify the model's resilience and efficacy in diagnosing different stages of DR.

The algorithm integrates data from various omics sources, such as DNA sequences, RNA transcripts, protein levels, and metabolite concentrations (*Huang, Shu & Liang, 2024*; *Sun et al., 2015*). Virtualising network functions and automating their orchestration ensures demanding applications can run efficiently with minimal delay while controlling operational costs (*Sun et al., 2018c*; *Shang & Luo, 2021*). It ensures that the destination prediction remains accurate and helpful while protecting the user's privacy (*Jiang et al., 2021*; *Shen et al., 2022*), and enhances intelligent city initiatives by providing data-driven insights for improving urban mobility and sustainability (*Xiao et al., 2021*; *Sun et al., 2018b*). The ARIMA model is used to analyse historical and real-time data to predict future traffic conditions and vehicle movements (*Sun et al., 2018a*; *Gu et al., 2024*). Considering the context of the conversation, including the relationship between interlocutors and the topic of discussion (*Ding et al., 2023*; *Pan et al., 2023*). Analysing how cultural backgrounds influence the perception of the game's themes, such as struggle, perseverance, and mortality (*Pan et al., 2024*; *Liu et al., 2024a*). This involves implementing and practically using VR in various educational settings to achieve better learning results (*Marougkas et al., 2023*; *Zhu, 2023*). The primary goal of surgery is to remove as much of the tumour as possible while preserving heart function (*Jiang & Yan, 2021*; *Liu et al., 2024b*). The model works in both domains to iteratively refine the image, leveraging the strengths of each domain for better reconstruction (*Li, 2024*; *Yang et al., 2024*). Applying algorithms (*e.g.*, K-means, hierarchical clustering) to group users into clusters based on the collected data (*Ban et al., 2023*; *Dang et al., 2023*). It uses convolutional layers that apply filters to input data to extract hierarchical features, making it practical for capturing spatial correlations (*Li et al., 2020*; *Qin et al., 2024*).

# PROPOSED KERNAL RANDOM FOREST WITH BLACK HOLE METHODOLOGY

The proposed methodology, KRF-BHO, aims to enhance the accuracy and efficiency of heart disease prediction by leveraging advanced feature selection and machine learning techniques. The methodology begins with multi-sensor data fusion, where sensor signals and echocardiogram data are collected and stored in a cloud-based system. The data undergoes extensive pre-processing steps, including imputation of missing values using K-nearest neighbors (KNN), normalisation, removal of redundant features, Z-score standardisation, and binary encoding. Feature selection is then performed using the Kernel Random Forest algorithm, which constructs multiple decision trees to identify significant features.

Figure 1 shows the architecture of the proposed work KRF-BHO.

BHOA further refines feature selection by mimicking the gravitational pull of a black hole, optimising the dataset for the most relevant features. The selected features are then classified using the XGBoost algorithm, which employs gradient boosting and regularisation to enhance predictive performance. The dataset is split into training and testing sets, with 5-fold cross-validation applied to ensure robust model evaluation. The proposed KRF-BHO methodology significantly improves prediction accuracy and computational efficiency, providing a reliable and scalable solution for heart disease diagnosis in clinical settings.

## Multi sensor data fusion

Sensor signal values are collected from various wearable sensor devices affixed to the patient's body, with Echocardiogram data collected under the supervision of a physician and fused and sent to the system through Bluetooth. In this work, the sensor signal values obtained from patients relate to heart disease. In the heart muscle, an ECG sensor signal detects the direction of electrical impulses. The range of abnormal heart rate is beyond 100 beats per minute. If the heart rate per minute is less than 60, it is termed Bradycardia and a heart rate greater than 100 beats per minute is called tachycardia. If the PR interval is more significant than 0.2, then the block is in the atrioventricular node. A TMP117 digital temperature sensor with Bluetooth connectivity detects the body's temperature. The normal body temperature is 98.6 °F (37 °C), and the abnormal body temperature is more significant than 100.4 °F (38 °C). A sensor (Honeywell's 26 PC SMT) is used to access the blood pressure. The normal blood pressure is below 120 in the systolic range and below 80 in the diastolic range (120/80). Elevated BP is a systolic number of more than 120 and/or a diastolic value below 80. For high BP, the systolic range is between 130 and 139; the diastolic range is between 81 and 9. The oximeter estimates the $SPO_2$ in the range of 70–99 percent. The oximeter ranges from 95 to 100 per cent in oxygen saturation for healthy people. It is below 95 per cent of the pulse oximeter for abnormal conditions. This fusion of multi-sensor values is stored as binary, and each value is separated by a comma in .csv file format. All these input files are kept in cloud storage. To calculate heart illness, these fused data are classified as binary to express the two class values: 1 indicates heart disease, and 0 means normal.

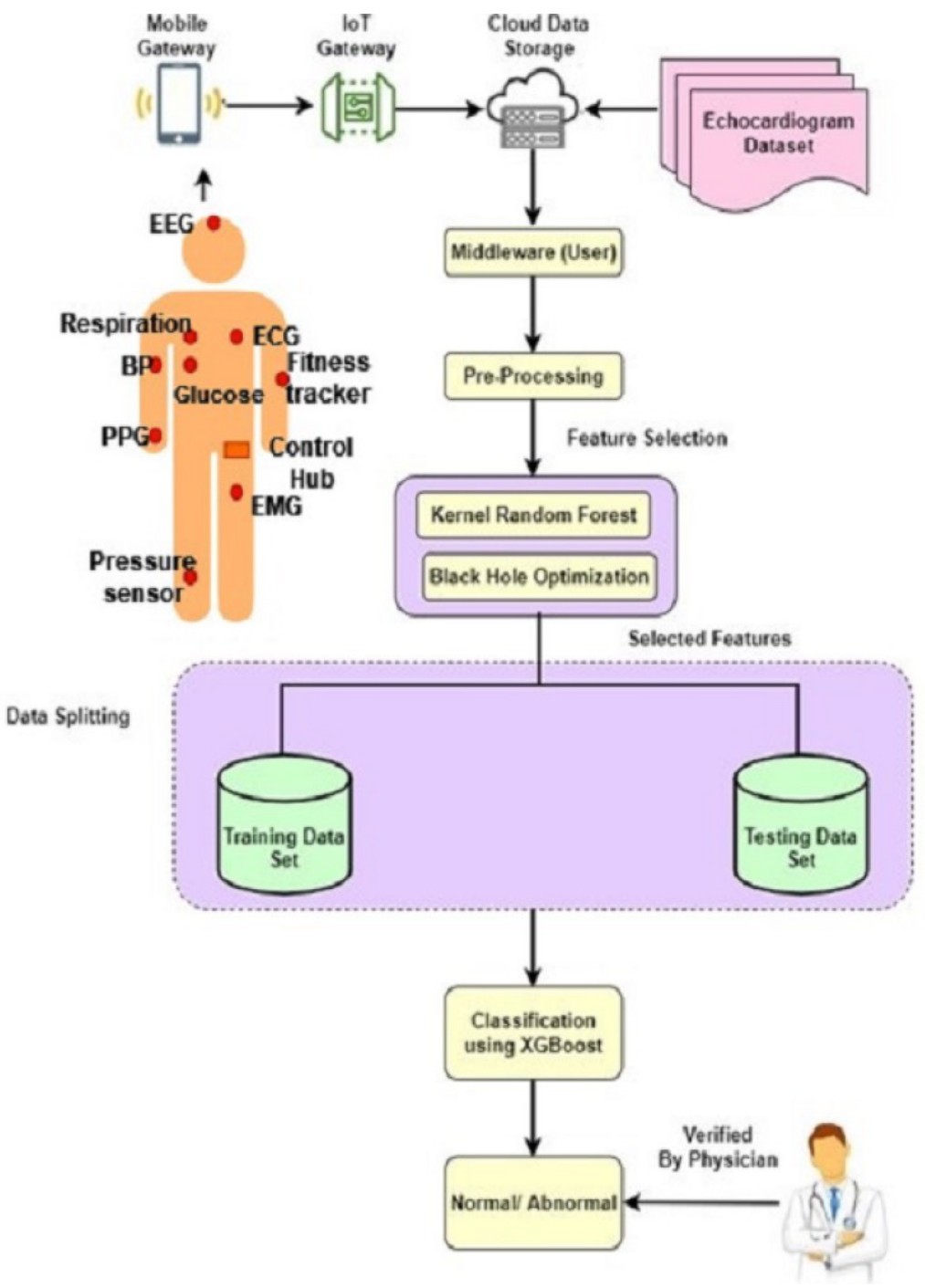

**Figure 1  Architecture of proposed model.**

## Data set description

The Cleveland dataset, stored at UCI, is used for AIoMT-based cardiac disease prediction. Sensors such as thermometers, electrocardiograms, pulse oximeters, and blood pressure

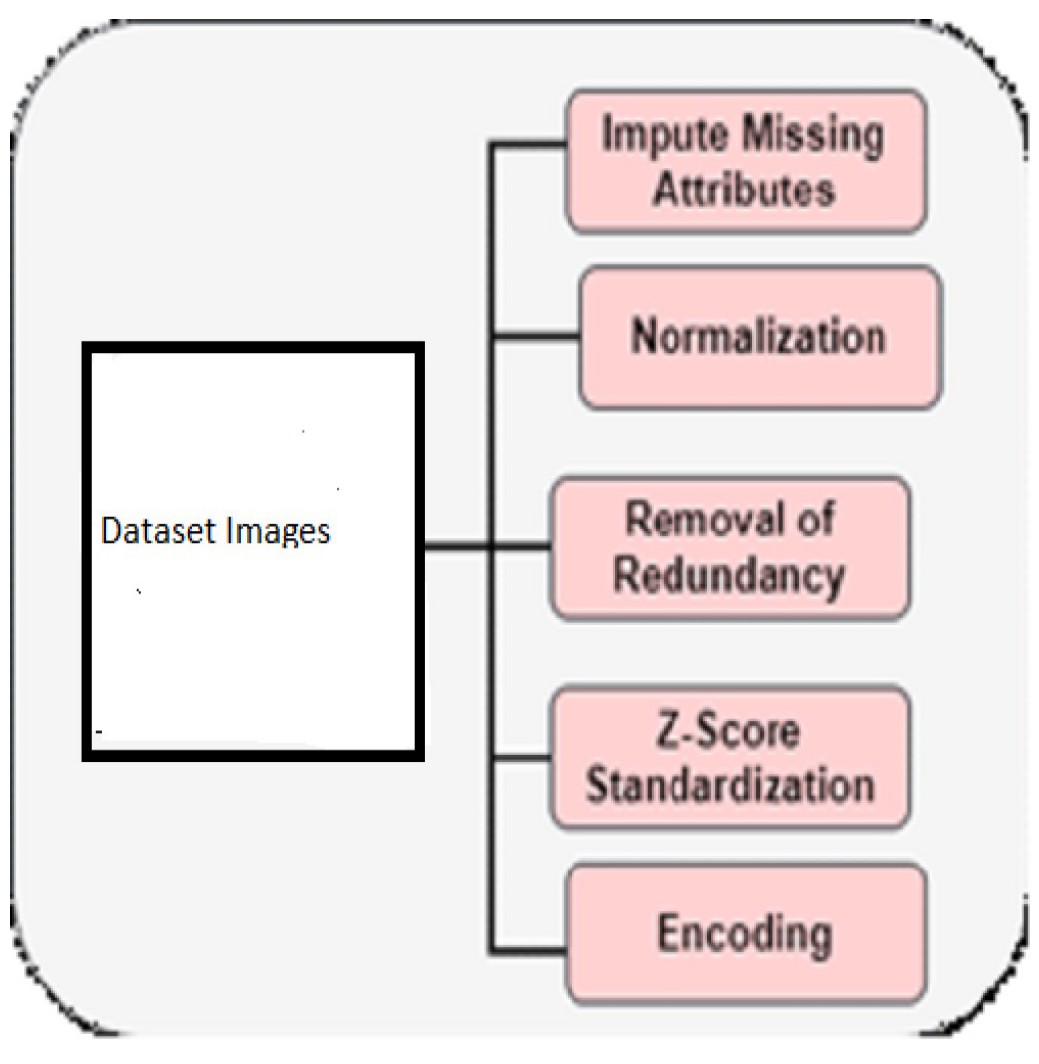

**Figure 2  Pre-processing.**

monitors contribute to the attribute collection process. Electrocardiogram (ECG), heart rate, blood pressure, and temperature readings are all produced by these sensor equipment. Using Internet of Things (IoT) technology, the data was captured and stored in the cloud. There is one intended output column, thirteen features, and thirty-five records. The desired output column has two categories: normal and cardiac disorders. Both the training and testing phases of the model make use of this dataset. To improve the quality and accuracy, pre-processing, which handles the missing attributes, removal of redundancy, normalisation, and so on, is needed. Figure 2 describes the pre-processing of predicting heart disease (*Alasadi & Bhaya, 2017*).

### Impute missing attributes
After assessing the patient's age, various sensor values from wearable devices, cholesterol, and so on, For some patients, some attribute values (sensor values) are missing. Adding

the missing attributes by using KNN. Identify the disappeared attribute by calculating the similarity or matching attributes in the dataset (*Jonsson & Wohlin, 2004*). To detect the matching process by using Euclidean distance is defined by:

$$D(P,Q) = \sqrt{\sum_{i=1}^{n}(P_i - Q_i)^2}  \tag{1}$$

where $P_i$ is a known value and $Q_i$ is a prediction value.

### Normalization

It converts the feature numeric values into new numbers based on minimum and maximum feature value (*Gökhan, Güzeller & Eser, 2019*) by using Eq. (2)

$$\overline{P} = \frac{P - min}{max - min}  \tag{2}$$

where *min* and *max* is the minimum and maximum value of selected feature $\overline{P}$ is the value after applying normalisation and $P$ is the selected feature numeric value.

### Removal of redundancy

It minimises the feature data set and improves accuracy by removing unrelated feature values from the data set. It removes irrelevant features based on the collected sensor signals and depending upon the patient's health issue.

### Z-score standardization

It converts the feature numeric values into new numbers using standard deviation and feature means (*Prasad, 2015*) by using Eq. (3).

$$\overline{P} = \frac{P - \mu}{\sigma}  \tag{3}$$

$\overline{P}$ is an output value after applying standard deviation and mean, and $P$ is the selected input value of the feature.

### Encoding

The collected feature values are categorised into two classes: 0 represents absence and 1 for presence in the new feature numeric value (*Potdar, Pardawala & Pai, 2017*).

## Proposed feature selection

For feature selection, a hybrid kernel random forest technique with the Black Hole Optimization algorithm (KRF-BHO) is used. Predicting heart disease involves collecting input heart-related data, sensor signal values, and echocardiogram images.

The KRF algorithm starts by transforming the original feature space into a higher-dimensional space using random Fourier features (RFF), which allows the Random Forest classifier to better capture non-linear relationships common in complex medical datasets. Each feature vector is mapped using the RBF kernel approximation, and a Random Forest classifier, consisting of multiple decision trees trained on bootstrap samples, is then trained on the transformed data. The importance of each feature is calculated based on its frequency

in splitting nodes in the trees, with more important features contributing significantly to reducing impurity (*e.g.*, Gini impurity or entropy). This process results in a ranked list of features, reducing the dataset's dimensionality while retaining the most informative features.

The parameters for Kernel Random Forest (KRF) and Black Hole Optimization algorithm (BHOA) were chosen through empirical testing, domain expertise, and systematic tuning to maximise model performance. For KRF, the RBF kernel was selected due to its effectiveness in handling non-linear relationships in medical data, with the gamma parameter set to 1.0 to balance bias and variance. The number of trees (estimators) was set to 100, based on experiments that showed this provided a good trade-off between performance and computational cost, while the maximum tree depth was set to 10 to capture complex patterns without overfitting. For BHOA, the population size (number of stars) was set to 30, and the algorithm was run for 100 iterations, parameters that were determined to offer a thorough exploration of the feature space while maintaining computational feasibility. These carefully selected parameters ensured that both the feature selection process and the final model performance were optimized.

### Kernel Random Forest

Let $P = \{P_1, P_2, \ldots, P_N\}$ are set of random variables and $Q = \{Q_1, Q_2, \ldots, Q_N\}$ are set of responses. A function $f(p) = \mathbb{E}[Q|P = p]$ predicts the response of random variable $P$. In the heart disease dataset $D = \{(P_1, Q_1), \ldots, (P_n, Q_n)\}$ of the range of values between $[0, 1]^d \times \mathbb{R}$ is independent pair of values in the form of $(P, Q)$, where $\mathbb{E}[Q^2] < \infty$. Compute the infinite random forest $f_{\infty, n}[0, 1]^d \rightarrow \mathbb{R}$ of $f$, of the dataset $D$. The collection of $T$ random trees, and the predicted value of $n$th tree then $p$ is $f_n(p, \Delta_j)$; the independent random variables for the data set $D$ are $\Delta_1, \Delta_2, \ldots, \Delta_n$. The outputs from every tree indecently are gathered to form finite forest, and it is defined by:

$$f_{p,n}(p, \Delta_1, \Delta_2, \ldots, \Delta_n) = \frac{1}{T} \sum_{j=1}^{n} \left( \sum_{i=1}^{p} \frac{Q_i \mathbb{I}_{p_{i \in \emptyset_n}}(p, \Delta_j)}{\tau_n(p, \Delta_j)} \right) \tag{4}$$

Algorithm 1 describes the randomness of data points from the dataset $D$ are selected. The probability of randomly choosing two fixed data points from the terminal nodes. It will give an interpretation of kernel-based random forest. Observe the maximum weight for the selected region data point in the forest, creating a rough set of estimates. In particular data regions, it may not contribute to the classifier which produces the prediction error. This prediction error introduces the sparse portion and weight problem in the forest by using a random forest-based kernel. For all $p \in [0, 1]^t$. In Eq. (4) define the data points of the region which contains $P$ and $Q_i$. If the region does not contain the data point $P$, it does not contribute to the prediction error.

### Black Hole Optimization algorithm (BHOA)

The black hole activity is in the actual galaxy. In an original space environment, a source is a black hole used for massive density through the attraction of gravitational force. It includes more gravitational forces which stars are attracted around it. In space, a block

---

**Algorithm 1** Kernel Random Forest Algorithm

---

**Require:** Values in the dataset $D$ in the form of $(P, Q)$ pairs and the number of trees in the forest $T, f \in \{1, 2, \ldots, n\}, p \in [0, 1]^t$

**Ensure:** Prediction of kernel random forest at $p$

1: **Step 1:** For each $k \in M$ do
2: **Step 2:** Select random points $\tau_n$ from the dataset $D$.
3: **Step 3:** For all $j \in \tau_n$, set $\rho_j = \emptyset, \rho_{ou} = [0, 1]^t$
4: **Step 4:** Let $\eta_{ver} = 1; \psi = 0$ //$\eta$ isthe number of vertices and $\psi$ is the level of tree
5: **Step 5:** While $\eta_{ver} < \tau_n$ do
6: **Step 6:** If $\psi \neq \emptyset$ then
7: **Step 7:** $\rho \leftarrow data\ point\ p$
8: **Step 8:** If $\sum \rho = 1$ then
9: **Step 9:** $\rho_\varphi \leftarrow \rho_\varphi \cup \rho$
10: **Step 10:** Else
11: **Step 11:** Split the data point in the set $\rho$ into $\rho_m, \rho_n$
12: **Step 12:** $\rho_{\varphi+1} \leftarrow \rho_{\varphi+1} \cup \rho_m \cup \rho_n$.
13: **Step 13:** $\eta_{ver} \leftarrow \eta_{ver} + 1$
14: **Step 14:** End If
15: **Step 15:** Else
16: **Step 16:** $\varphi \leftarrow \varphi + 1$
17: **Step 17:** End If
18: **Step 18:** End While
19: **Step 19:** Evaluate $f(p, \Delta_i, D)$ // Local prediction for $p$
20: **Step 20:** Evaluate $f_{p,n}(p, \Delta_1, \Delta_2, \ldots, \Delta_n, D)$ // Global prediction for $p$
21: **Step 21:** Return

---

hoke is created when stars collapse the galaxy. Because nothing is absorbed, it is called black. In the galaxy, the outer space black hole is classified as a sphere-shaped bond and treated as an event horizon. The event horizon radius is called a Schwarzschildradius. Due to heavy gravitational force in the black hole, the star comes to an end at the event horizon, which absorbs the black hole and vanishes. In the event horizon, the amount of light speed is treated as escape velocity; therefore, nothing can go away from the inside of the event horizon. In the black hole optimization technique, Euclidean distance is measured between the star and the black hole. If the distance is less in radius of Schwarzschild, then replace it with a new star in its search space at an arbitrary position. The position of the new star is based on the lowest cost in the black hole. For changing the position of the star by using the Hyperbolic Tangent function. In the search space, stars are positioned randomly. Each star has its own fitness value, and it is evaluated by using the evaluation function of fitness. The calculated fitness values are based on the accuracy rate of the classifier. The optimal best solution is referred to as a black hole. Based on the best fitness value star selects the black hole. The position of the star gets modified if it has a minimum number of features. Repeat the process until it improves its accuracy rate. This BHOA is implemented by using

three phases namely populating stars, selection of Black Hole and updating position of stars.

*Populating stars.* Let us assign the maximum number of iterations. Initialize the population of the star and randomly select the attributes, populate the stars. The populating star algorithm is given below:

---

**Algorithm 2** Populating Stars

---

**Require:** Star's Dataset $S\_D$, Number of Stars, Max iterations

**Ensure:** Black hole, Fitness value of Black hole

  1: **Step 1:** Initialize population of the star
  2: **Step 2:** Begin
  3: **for** $i = 1$ to *num_stars* **do**
  4:     **for** $j = 1$ to *no_attributes* **do**
  5:         Generate $rnd[i][j] = 0$ or 1
  6:         **if** $rnd[i][j] == 1$ **then**
  7:             Add attribute to $S\_D$
  8:         **end if**
  9:     **end for**
10: **end for**

---

In the Algorithm 2, the maximum number of iterations is considered as 600. The first step of the black hole optimization algorithm is to initialise the star population. If the random value is 1 then add the attribute in the star's data set. For better performance choose the number of stars.

*Selection of black hole.* By using the classification techniques fitness value is calculated for star. The optimized star value is considered as black hole. In the black hole, if stars have the same optimal value, then the selection is based on the replacement of a star which has fewer features minimum. The algorithmic procedure of selection of a black hole is given below:

*Updating star's position.* By using a tangent function with a threshold value of 0.7, update the star's position. For the selection of features use the Hyperbolic tangent function, which returns zero or one. The position of the star is updated by using Eqs. (5) & (6).

$$w(s_{id})(b+1) = abs(tanh(s_{id}(b+1))) \tag{5}$$

$$s_{id}(b+1) = \begin{cases} 1 \ if \ \ w(s_{id}(b+1)) > rnd \\ 0 \hspace{2.5cm} otherwise \end{cases} \tag{6}$$

This algorithm gives the optimal features of the heart image data set, which reduces the training time and improves its accuracy. The most relevant and optimized features are

---

**Algorithm 3** Selection of Black Hole

---

**Require:** Number of Stars, Max iterations

**Ensure:** Black hole

  1: **for** $i = 1$ to max_iter **do**

  2:      **for** $j = 1$ to No_stars **do**

  3:          Evaluate fitness value of star using RandomForest Classifier

  4:          **if** $\big(\mathrm{fit}(X[j]) > \mathrm{fit}(X[black\_hole])\big)$ **then**

  5:              $X[black\_hole] = X[j]$

  6:          **else if** $\big(\mathrm{fit}(X[j]) == \mathrm{fit}(X[black\_hole])\big)$ **and** $\big(|X[j]| < |X[black\_hole]|\big)$ **then**

  7:              $X[black\_hole] = X[j]$

  8:          **end if**

  9:          **if** $X[j]$ is a duplicate of *black_hole* **then**

10:              Replace $X[j]$ by new star.

11:          **end if**

12:      **end for**

13: **end for**

---

**Algorithm 4** Updating Star's Position

---

**Require:** Number of Stars, Number of Features

**Ensure:** Star Position

  1: **for** $i = 1$ to no_stars **do**

  2:      **for** $k = 1$ to no_features **do**

  3:          $X[i][k] = X[i][k] + 0.7 \times (X[black\_hole][k] - X[i][k])$

  4:          **if** $(|\tanh(X[i][k])| > \mathrm{rnd})$ **then**

  5:              $X[i][k] = 1$

  6:          **else**

  7:              $X[i][k] = 0$

  8:          **end if**

  9:      **end for**

10: **end for**

---

type of chest pain, Thallium scan, exercise-induced angina *etc*. The working flow of BHO is given in Fig. 3.

## Data splitting

The dataset used in this proposed work is divided into 80% training set, a 20% as the testing set, and also five-fold cross-validation in the training dataset.

    In our Python implementation, BHOA was used to refine the feature selection process. The algorithm starts by initializing a population of potential solutions, referred to as stars, where each star represents a subset of features randomly selected from the training data. The algorithm iteratively evaluates the fitness of each star by scoring the performance of a pre-trained KRF model on the subset of features represented by that star. The star with the highest fitness score is designated as the "best star", analogous to the black hole

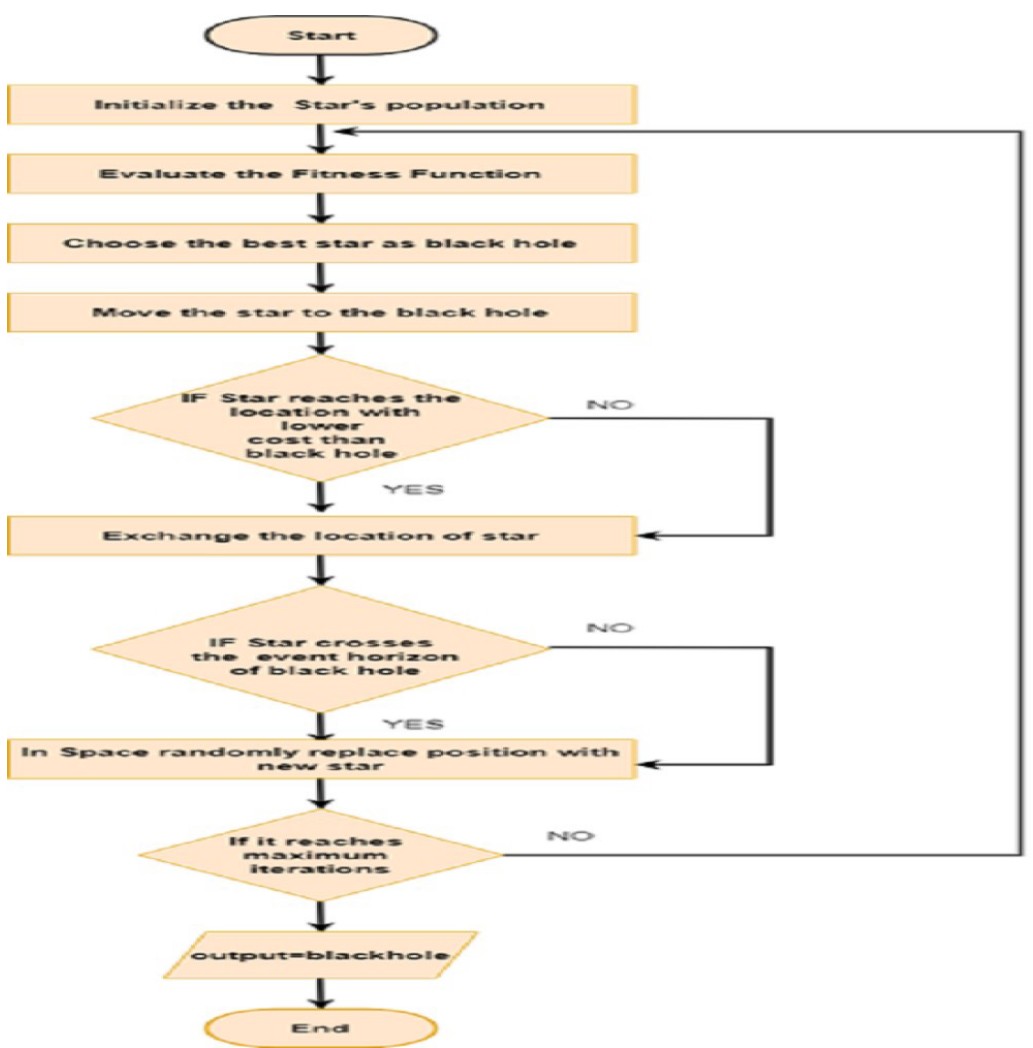

**Figure 3** Flow diagram of BHO.

in the algorithm. Each iteration of the BHOA process is a step towards a more refined feature subset. The fitness of all stars is evaluated, and if any star surpasses the current best star, it becomes the new best star. The positions of the stars are then updated, with each star's features adjusted to resemble those of the best star more closely. This adjustment is probabilistic, with each feature having a 50%.

Chance of being inherited from the best star. This iterative process continues for a specified number of iterations, ensuring that the feature subset evolves towards an optimal configuration. Finally, the algorithm outputs the best-performing subset of features, which are then used to create optimised training and testing datasets. This refined feature set enhances the performance of the subsequent machine-learning model, leading to more accurate and efficient predictions. The combination of KRF for initial feature selection and BHOA for optimisation provides a robust framework for identifying the most relevant features in the dataset.

## Classification using XGBoost

The eXtreme gradient boosting (XGBoost) is used to measure the performance of the proposed feature selection as the KRF-BHO algorithm. It is a supervised learning technique of both regression and classification. The working principle of XGBoost is gradient boosting. Implementation of gradient boosting towards column sampling, loss function, and regularization. The main aim of XGBoost is to analyse the error residuals to find out the prediction outcomes and also help to reduce loss scores. The regularization with loss function are shown in Eqs. (7) & (8):

$$loss(\emptyset) = \sum_i I(\widehat{x_i}, \ x_i) + \sum_w \Omega(f_w) \tag{7}$$

$$\Omega(f) = \Upsilon P + \frac{1}{2}\lambda \|TS\|^2. \tag{8}$$

Here '$loss$' is the loss function which evaluates the target and prediction models difference. The term ?? is regularization with the complexity value with the number of leaves in the tree $TS$. $f_w$ is the leaf weight. $\Upsilon$ is the threshold value and $\lambda$ is the learning rate which adopt the over-fitting problem.

---

**Algorithm 5** Regression and Classification Algorithm

---

1: Initialize the target prediction function from the KRF-BHO
2: **for** $i = 1$ to $m$ **do**
3:   Evaluate loss function and regularization using Eqs. (7) & (8)
4:   To avoid over-fitting problem, fit the model using Eq. (7)
5:   Select the step size of gradient descent
6:   Update the estimation function value
7: **end for**
8: Obtain the regression model
9: Classify the label for predicting the heart disease

---

The KRF-BHO model generates the appropriate feature selection. ThisXGBoost classifier model is adopted for learning the training and testing dataset and classifies the condition of heart disease as normal and abnormal.

In our research, we implemented XGBoost for image-based heart disease prediction by following a detailed and structured approach. We began by loading and preprocessing the images organized in directories corresponding to their class labels. Each image was resized to a uniform dimension of 64 × 64 pixels to ensure consistency across the dataset. The resized images were then converted to numpy arrays and flattened into 1D vectors, creating feature vectors suitable for machine learning models. Our investigation used XGBoost to predict heart disease based on images. We followed a meticulous and organised approach throughout our study. We started by loading and preprocessing the images, which were neatly organised in directories based on their class labels. Every image was adjusted to a consistent size of 64x64 pixels to maintain uniformity throughout the dataset. After

resizing, the images were transformed into numpy arrays and flattened into 1D vectors. This process resulted in feature vectors that are well-suited for machine learning models.

To make the training process easier, we converted the class labels into integers using a label encoding technique. This step ensured that the categorical labels were converted into a format compatible with the XGBoost algorithm. The dataset was divided into training and testing sets using an 80–20 split, which ensures a reliable foundation for evaluating the model. To train the model, we began by initialising an XGBoost classifier and then optimised it using the 'logloss' evaluation metric. The model underwent extensive training using the available data, enabling it to accurately distinguish between various types of heart disease by analysing the image characteristics. After the model was trained, its performance was assessed on the test set using accuracy as the main metric for evaluation. In addition, we used feature importance visualisation to gain insights into which features (pixels) had the greatest impact on the model's predictions. This visualisation was instrumental in better understanding the key areas in the images that are crucial for classifying heart disease. It offered valuable insights into the patterns that the model recognises. In summary, using XGBoost for image-based heart disease prediction showcased substantial enhancements in prediction accuracy and computational efficiency, providing a dependable solution for clinical diagnosis. By implementing a comprehensive preprocessing technique, conducting rigorous model training, and conducting meticulous evaluation, we could ensure our approach's effectiveness and reproducibility.

## RESULT & DISCUSSION

The dataset used for this proposed study is designed to be used as a training set with 80% of the data and a 20% testing set. The training dataset also underwent 5-fold cross-validation.

### Data set description

Pre-processing is needed to improve the quality and accuracy. This process handles the missing attributes, redundancy removal, normalisation, *etc*. The dataset used to predict heart disease based on AIoMT is the Cleveland dataset from the UCI repository. The attributes are collected from sensor devices like temperature sensors, ECGs, pulse oximeters, and temperature and blood pressure sensors. These sensor devices generate ECG data, heart rate, blood pressure, and body temperature. The data were collected and saved in the cloud using IoT technology. It has 350 records, 13 features, and one targeted output column. The target output column includes two classes: 1 indicates heart diseases, and 0 indicates normal. This dataset is used in both training and testing the model. Table 2 is the description of Cleveland dataset (*Khan & Algarni, 2020*).

The echocardiogram description in image dataset is shown in Table 3 with appropriate attributes.

### Statistical metric measures

The statistical metric measures used in the prediction of heart disease are Kappa statistic, classification accuracy, and errors in heart disease.

**Table 2  Description of features in the Cleveland dataset.**

| Features | Type | Description |
|---|---|---|
| Age | Continuous | Age in years |
| Sex | Discrete | 1 = Male; 0 = Female |
| Chol | Continuous | Serum cholesterol in mg/dL |
| Cp | Discrete | Types of chest pain (0: Typical Angina, 1: Atypical, 2: Non-anginal pain, 3: Asymptomatic) |
| Trestbps | Continuous | Resting blood pressure (mmHg) |
| Fbs | Discrete | Fasting blood sugar ¿120 mg/dL; True = 1, False = 0 |
| Exang | Discrete | Exercise-induced angina (1 = Yes, 0 = No) |
| Oldpeak ST | Continuous | ST depression induced by exercise relative to rest |
| Restecg | Discrete | Resting electrocardiographic results (0: Normal, 1: Abnormal) |
| Thalach | Continuous | Maximum heart rate achieved |
| Thal | Discrete | Thalassemia (0 = Normal, 1 = Fixed Defect, 2 = Reversible Defect) |
| Target output | Discrete | Diagnosis of heart disease (0: Abnormal, 1: Normal) |
| Slope | Discrete | Slope of the peak exercise ST segment (0: Upsloping, 1: Flat, 2: Downsloping) |

**Table 3  Features of the echocardiogram image dataset.**

| Feature | Description |
|---|---|
| Survival | Patient's survival; 0: Dead; 1: Alive |
| Age at heart disease | Age of patient when heart attack happened. |
| Fractional_shortening | The measurement of heart to eject a stroke volume. Lesser numbers were very abnormal. |
| Pericardial effusion | Fluids in heart: fluid means 1; no fluid means 0. |
| Epss | Septal separation of E-points, various contractility measurements. Majority abnormal numbers. |
| Lvdd | Left ventricular dimension on end-diastolic. It denotes the heart size at end-diastole. The size of the heart is large, is considered a sick heart. |
| Wall motion score | It measures the function of left ventricles are functioning. |
| Wall motion index | Wall motions score is divided by numbers of parts in the image of echocardiogram. In general, 12–13 segments were seen in the echocardiogram. These variables were used instead of the wall motion scores. |
| Name | Name of the patient. |

### Kappa statistic

These statistic measures are classified as the expected and observed heart disease, and it is formulated as:

$$\frac{Observed\ values - Expected\ Values}{1 - Expected\ Values} \tag{9}$$

The Kappa statistic value is the range between 0 to 1. If it is near 1, it will be agreed near the exact disease prediction; otherwise, there is no prediction.

**Table 4 Statistical metric measure.**

| Class | Input | Kappa | Accuracy |
|---|---|---|---|
| Normal | Multi-sensor data fusion | 0.7806 | 96.35 |
| | Cleveland Dataset | 0.8775 | 94.74 |
| Abnormal | Multi-sensor data fusion | 0.7252 | 97.34 |
| | Cleveland Dataset | 0.8956 | 96.22 |

### *Accuracy*

$$Accuracy = \frac{Correctly\ predicted\ class}{Total\ testing\ class} \times 100\% \tag{10}$$

Accuracy = (correctly predicted class / total testing class) ×100%

Table 4 shows that statistical metric measures for the KRF-BHO algorithm.

The observations of Table 4 shows that the proposed work KRF-BHO algorithm was tested in the aspects of kappa and accuracy metric measures based on the data collected from various wearable devices and the Cleveland dataset.

## Performance metric measures

These parametric metric measures are computed to predict heart disease based on the Cleveland dataset and the fusion of multi-sensor values in this proposed work KRF-BHO. This proposed work is compared with existing algorithms of KRF (*Muzammal et al., 2020*), and BHOA (*Rajadevi et al., 2021*). The classifier XGBoost.

**True positive (TP):** The heart patient is in a healthy condition and is also predicted as a healthy condition.

**False positive (FP):** The heart patient is healthy but predicts an abnormal healthy condition.

**True negative (TN):** The heart patient is in an unhealthy condition and predicted as an unhealthy condition.

**False negative (FN):** The heart patient is in an unhealthy condition, but predicts a healthy condition.

$$Specificity = \frac{TN}{TN + FP} \tag{11}$$

$$Accuracy = \frac{TP + TN}{TP + TN + FP + FN} \tag{12}$$

### *Precision*

It is called positive predictive value (PPV). It evaluates true positive for all positive values by using

$$Precision = \frac{TP}{TP + FP} \tag{13}$$

**Table 5  Metric measures of precision and recall.**

| Algorithm | Training | | | | Testing | | | |
| --- | --- | --- | --- | --- | --- | --- | --- | --- |
| | Multi sensor data fusion | | Cleveland dataset | | Multi sensor data fusion | | Cleveland dataset | |
| | Precision | Recall | Precision | Recall | Precision | Recall | Precision | Recall |
| KRF | 76.1% | 78.6% | 78.3% | 81.4% | 78.8% | 79.3% | 76.6% | 72.4% |
| BHOA | 67.6% | 68.8% | 59.9% | 61.2% | 71.3% | 69.8% | 72.2% | 69.4% |
| KRF-BHO (Proposed) | 92.5% | 91.3% | 94.1% | 93.6% | 94.3% | 92.7% | 92.6% | 91.1% |

### Recall

It evaluates true negatives for all negative values by using

$$Recall = \frac{TP}{TP + FN} \tag{14}$$

### F-score

$$F\text{-}score = 2 \times \frac{Precision \times Recall}{Precision + Recall} \tag{15}$$

From Table 5, the precision of the KRF-BHO algorithm is better than KRF (76.1%) and BHOA (67.6%) in the training dataset of multisensor data fusion, and similarly, KRF-BHO outperforms other algorithms in the Cleveland dataset with precision of 94.1%. In the testing data set, the precision of the KRF-BHO algorithm is better than KRF (78.8%) and BHOA (71.3%) of multi-sensor data fusion, and similarly, KRF-BHO outperforms other algorithms in the Cleveland dataset with the precision of 94.3%. For the recall of KRF-BHO algorithm is better than KRF (78.6%) and BHOA (68.8%) in the training dataset of multi-sensor data fusion, and similarly, KRF-BHO outperforms other algorithms in the Cleveland dataset with the precision of 93.6%. In the testing data set, the recall rate in the KRF-BHO algorithm is better than KRF (79.3%) and BHOA (69.8%) of multi-sensor data fusion. Similarly, KRF-BHO outperforms other algorithms in the Cleveland dataset with a precision of 91.1%. Figure 4 shows that the training phase of Specificity and F1-score of multi-sensor data fusion and Cleveland dataset are used.

Using medical signal data gathered by multi-sensors, the proposed KRF-BHO technique in the training phase, 93.1% specificity, 91.3% in F1-score and for the Cleveland dataset, 94.1% specificity, 91.3% in F1-score 93.6% were obtained. Figure 5 shows that the testing phase of Specificity and F1-score of multisensor data fusion and Cleveland dataset are used.

Using medical signal data gathered by multi-sensors, the proposed KRF-BHO technique in the testing phase, 94.3% specificity, 92.7% in F1-score and for the Cleveland dataset 92.6% specificity, 91.3% in F1-score 91.1% were obtained. The performance of accuracy, precision, recall, specificity and F1-score and comparison of the medical image classification model using CheXNet and VGG-19 (*Basheer, Alluhaidan & Bivi, 2021*). Table 6 shows the classification of heart disease using various classifiers in the data collected from multiple wearable sensor devices (multi-sensor data fusion) data sets.

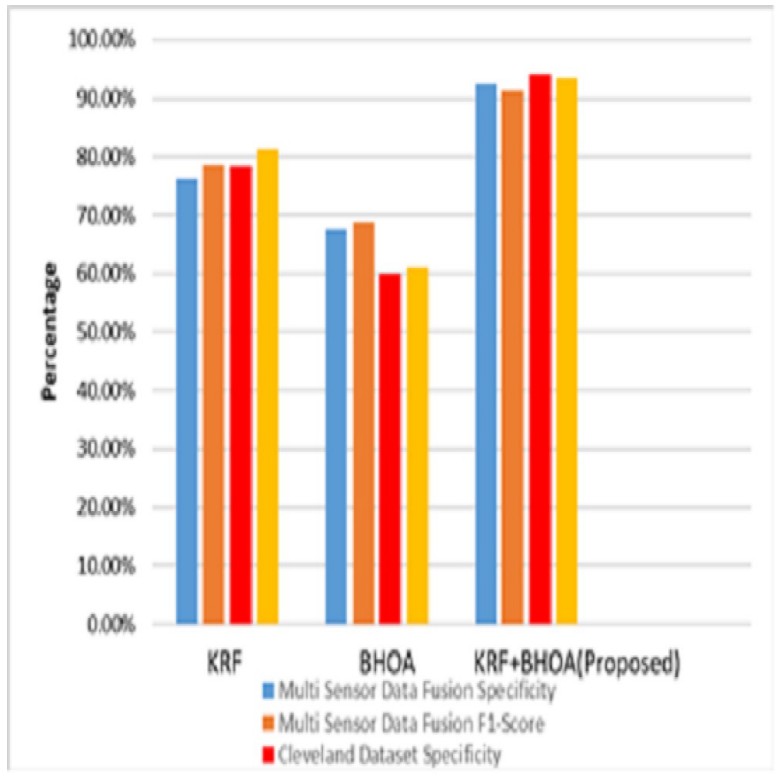

**Figure 4  Sensitivity & F1-score (Training).**

**Table 6  Classifier technique in the multi-sensor data fusion dataset.**

| Algorithm | Precision | Recall | Specificity | F1-score | Accuracy |
|---|---|---|---|---|---|
| VGG-19 *Basheer, Alluhaidan & Bivi (2021)* | 93.12 | 94.56 | 93.22 | 95.85 | 89.12 |
| CheXNet *Basheer, Alluhaidan & Bivi (2021)* | 91.32 | 90.81 | 89.77 | 91.33 | 88.89 |
| XGBoost *Rajadevi et al. (2021)* | 94.56 | 94.11 | 92.77 | 94.23 | 90.12 |
| KRF-BHO with XGBoost (proposed) | 95.61 | 96.66 | 95.12 | 97.34 | 95.45 |

In the observation of Table 6, the classifier XGBoost with proposed work (KRF-BHO) produces better results in 95.61% precision, 96.66% recall, 95.12% specificity, 97.34% F1-score and 95.45% in accuracy. Table 7 shows the classification of heart disease by using various classifiers in the data collected from various wearable sensor devices (multi-sensor data fusion) data sets.

In the observation of Table 7, the classifier XGBoost with proposed work (KRF-BHO) produces a better results in 96.69% precision, 95.85% recall, 96.72% specificity, 96.78% F1-score and 96.4% in accuracy. Table 8 shows the classifier's performance in the aspect of accuracy after features are selected in both training and testing multi-sensor data fusion data set.

In the observation of Table 8, after selecting features, our proposed work outperforms better results in the accuracy analysis. Table 9 shows the classifier's performance in the

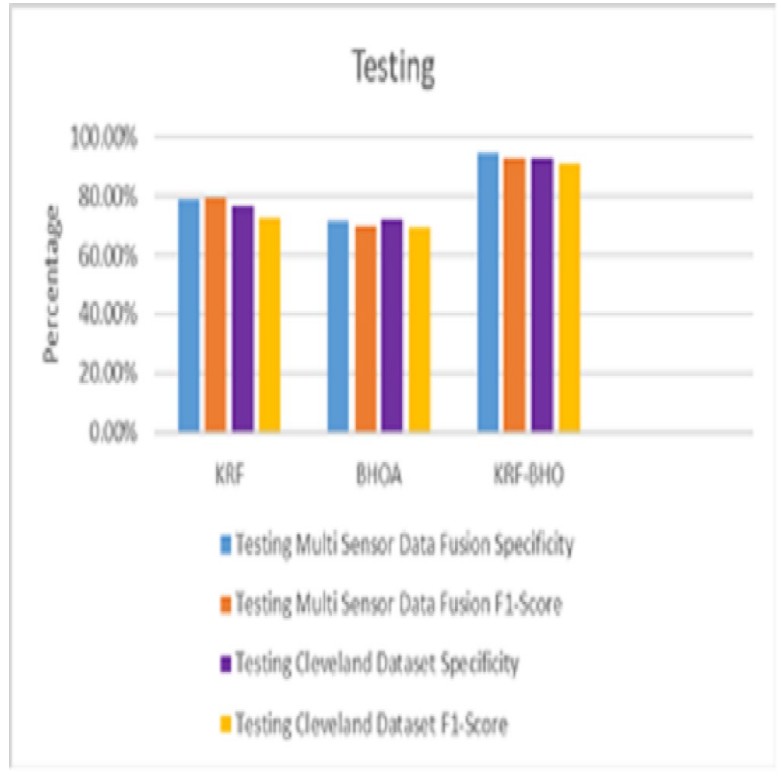

**Figure 5**    **Sensitivity & F1-score (Testing).**

**Table 7**    **Classifier technique in the Cleveland dataset.**

| Algorithm | Precision | Recall | Specificity | F1-score | Accuracy |
|---|---|---|---|---|---|
| VGG-19 | 92.55 | 93.67 | 92.89 | 94.58 | 90.12 |
| CheXNet | 90.78 | 91.81 | 89.89 | 90.53 | 89.11 |
| XGBoost | 95.23 | 94.78 | 93.17 | 94.11 | 90.34 |
| KRF-BHO with XGBoost (Proposed) | 96.69 | 95.85 | 96.72 | 96.78 | 96.45 |

**Table 8**    **Multi sensor data fusion in accuracy (feature selected).**

| Algorithm | Selected features | Accuracy in training | Accuracy in testing |
|---|---|---|---|
| KRF | 12 | 89.12 | 90.45 |
| BHOA | 9 | 88.89 | 89.53 |
| KRF-BHO with XGBoost Classifier | 6 | 94.12 | 95.89 |

aspect of accuracy after features are selected in both training and testing multi-sensor data fusion data set.

In the observation of Table 9, after selecting features, our proposed work outperforms better results in the accuracy analysis with fewer features from the heart disease dataset. Figure 6 shows the computation time of various techniques in predicting heart disease.

**Table 9  Cleveland dataset in accuracy (feature selected).**

| Algorithm | Selected features | Accuracy in training | Accuracy in testing |
|---|---|---|---|
| KRF | 12 | 90.89 | 91.52 |
| BHOA | 9 | 89.81 | 90.11 |
| KRF-BHO with XGBoost Classifier | 6 | 95.78 | 96.21 |

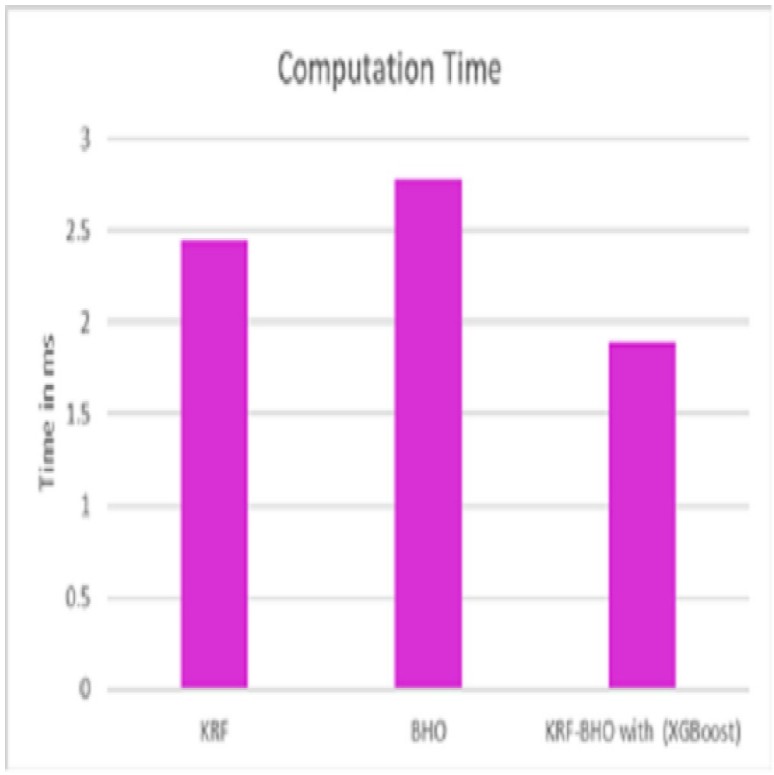

**Figure 6  Computation time.**

Given the high accuracy reported in the results, several measures were implemented to prevent overfitting during the model training process. Overfitting occurs when a model learns the underlying patterns and noise in the training data, leading to poor generalisation of new, unseen data.

To prevent overfitting during model training and ensure the high accuracy reported (95.45% for the multi-sensor data fusion dataset and 96.45% for the Cleveland dataset) was reliable, several measures were taken. Cross-validation with k-fold ($k = 5$) ensured robust performance evaluation, while regularisation techniques in XGBoost (lambda = 1, alpha = 0.5) penalised complex models. Early stopping was employed, halting training when the validation performance ceased improving after 10 rounds. The KRF-BHO methodology effectively selected the most relevant features, reducing data dimensionality. Hyperparameter tuning optimised parameters like learning rate (0.1) and max depth (6).

**Table 10  Metric measures of precision and recall.**

| Metric | Multi sensor data fusion | | Multi sensor data fusion | |
|---|---|---|---|---|
| | **Without KRF-BHO** | **With KRF-BHO** | **Without KRF-BHO** | **With KRF-BHO** |
| Accuracy (%) | 85.67 | 95.45 | 86.34 | 96.45 |
| Precision (%) | 84.23 | 95.61 | 85.12 | 96.69 |
| Recall (%) | 83.78 | 96.66 | 84.78% | 95.85 |
| Specificity (%) | 82.91 | 95.12 | 83.67 | 96.72 |
| F1-score (%) | 84.00 | 97.34 | 84.95 | 96.78 |
| Computational time (s) | 45.6 | 30.2 | 42.8 | 28.7 |

These steps collectively helped prevent overfitting, ensuring the model's accuracy was genuine and generalisable.

## KRF-BHO impact on overall model performance

The following tables compare the performance metrics of the heart disease prediction model with and without the KRF-BHO methodology, using XGBoost as the classifier. The metrics include accuracy, precision, recall, specificity, F1-score, and computational time for the multi-sensor data fusion and Cleveland datasets. Table 10 shows KRF-BHO's impact on overall model performance.

Table 10 illustrates the impact of the KRF-BHO methodology on the overall performance of heart disease prediction models, comparing results for the multi-sensor data fusion dataset and the Cleveland dataset. The results demonstrate significant improvements across all metrics when using the KRF-BHO approach. For the multi-sensor data fusion dataset, accuracy increased from 85.67% to 95.45%, precision from 84.23% to 95.61%, recall from 83.78% to 96.66%, specificity from 82.91% to 95.12%, and F1-score from 84.00% to 97.34%. Computational time also decreased from 45.6 seconds to 30.2 seconds, highlighting the efficiency gains of the proposed method. Similarly, for the Cleveland dataset, accuracy improved from 86.34% to 96.45%, precision from 85.12% to 96.69%, recall from 84.78% to 95.85%, specificity from 83.67% to 96.72%, and F1-score from 84.95% to 96.78%. Computational time was reduced from 42.8 seconds to 28.7 seconds. These enhancements illustrate the efficacy of the KRF-BHO methodology in selecting the most relevant features and optimising the model's performance. The substantial gains in precision, recall, and F1-score indicate a more reliable and accurate prediction capability, while the reduced computational time underscores the method's efficiency. This robust performance across both datasets demonstrates the KRF-BHO methodology's potential for widespread application in medical data analysis and heart disease prediction.

Table 11 presents the accuracy of the heart disease prediction model using different gamma and population size values for both the Multi-sensor Data Fusion and Cleveland datasets. The values illustrate how varying these parameters affects model performance.

The choice of KRF-BHO over existing methods is motivated by the specific strengths and advantages it offers in addressing the challenges of heart disease prediction. Traditional approaches often struggle with medical data's high-dimensional and complex nature, leading to suboptimal feature selection and model performance. KRF effectively addresses

**Table 11  Impact of different gamma and population size on accuracy.**

| Gamma | Population size | Multi-sensor data fusion accuracy (%) | Cleveland dataset accuracy (%) |
|---|---|---|---|
| 0.5 | 20 | 90.12 | 91.34 |
| 0.5 | 30 | 92.34 | 93.45 |
| 0.5 | 40 | 91.48 | 92.89 |
| 1.0 | 20 | 94.56 | 95.12 |
| 1.0 | 30 | 95.45 | 96.45 |
| 1.0 | 40 | 94.89 | 95.78 |
| 1.5 | 20 | 93.67 | 94.45 |
| 1.5 | 30 | 94.34 | 95.56 |
| 1.5 | 40 | 94.12 | 95.23 |

this issue by leveraging kernel methods to map the data into a higher-dimensional space where linear separability is enhanced. This capability is crucial for capturing the intricate patterns in multi-sensor medical data, making it a superior choice for initial feature selection. However, KRF alone may not always find the most optimal subset of features due to its inherent limitations in exploring the feature space comprehensively. This is where the BHOA comes into play. BHOA is inspired by the astrophysical phenomenon of black holes, which have a powerful gravitational pull that can attract surrounding stars (features) towards a central optimal point. This optimisation algorithm searches the feature space more thoroughly and avoids local optima, a common issue with traditional optimisation techniques. By combining KRF with BHOA, the proposed algorithm benefits from the robust initial feature selection of KRF and the powerful optimisation capabilities of BHOA. This hybrid approach selects the most relevant and significant features, enhancing the subsequent machine-learning model's predictive performance and computational efficiency.

Compared to more conventional approaches, the suggested KRF-BHO with XGBoost classifiers achieves better accuracy, efficiency, and computational time savings. These results highlight the promise of our method for accurate and rapid prediction of cardiac illness in clinical settings. Finally, we highlight the most important takeaways from this study and provide some suggestions for where the field may go from here.

## CONCLUSION

AIoMT-based heart disease prediction using kernel random forest with black hole classifier in XGBoost classifier was implemented. Medical data are collected from the patient's body using various wearable sensor devices and echocardiogram images from the dataset. The data set used in this proposed work was the Cleveland dataset, a multisensor data fusion data set. Accuracy in the training phase with multisensor data fusion data set of proposed work KRF-BHO with XGBoost classifier is 94.12%, and in the testing phase, the accuracy rate is 95.89%. Similarly for the Cleveland dataset, the proposed KRF-BHO with XGBoost classifier is 95.78%, the testing phase accuracy rate is 96.21%. Finally, this study shows that XGBoost and Kernel Random Forest, when combined, are the best classifiers for

predicting cardiac problems. Thanks to our method's ability to decrease computing time while improving prediction accuracy, it is feasible to use it in real-time clinical settings. Investigating the incorporation of more classifiers to further enhance performance and expanding the scope of this technology to handle bigger and more varied datasets are potential directions for future study.

### Funding

This work was supported by the Deanship of Scientific Research at King Khalid University through a large group Research Project under grant number (RGP2/86/45). Princess Nourah bint Abdulrahman University Researchers Supporting Project number (PNURSP2024R234), Princess Nourah bint Abdulrahman University, Riyadh, Saudi Arabia. Researchers Supporting Project number (RSPD2024R787), King Saud University, Riyadh, Saudi Arabia. This study is also funded by the Future University in Egypt (FUE). All the external funding or sources of support received during this study. The funders had no role in study design, data collection and analysis, decision to publish, or preparation of the manuscript.

### Grant Disclosures

The following grant information was disclosed by the authors:
The Deanship of Scientific Research at King Khalid University: RGP2/86/45.
Princess Nourah bint Abdulrahman University Researchers: PNURSP2024R234.
King Saud University, Riyadh, Saudi Arabia: RSPD2024R787.
The Future University in Egypt (FUE).

### Competing Interests

The authors declare there are no competing interests.

### Author Contributions

- Ala Saleh Alluhaidan conceived and designed the experiments, performed the computation work, prepared figures and/or tables, and approved the final draft.
- Mashael Maashi conceived and designed the experiments, analyzed the data, authored or reviewed drafts of the article, and approved the final draft.
- Noha Negm conceived and designed the experiments, analyzed the data, authored or reviewed drafts of the article, and approved the final draft.
- Shoayee Dlaim Alotaibi performed the experiments, performed the computation work, prepared figures and/or tables, and approved the final draft.
- Ibrahim R. Alzahrani performed the experiments, analyzed the data, performed the computation work, prepared figures and/or tables, authored or reviewed drafts of the article, and approved the final draft.
- Ahmed S. Salama performed the experiments, performed the computation work, authored or reviewed drafts of the article, and approved the final draft.

## Data Availability

The Heart Disease dataset is available at UCI: Janosi, A., Steinbrunn, W., Pfisterer, M., & Detrano, R. (1989). Heart Disease [Dataset]. UCI Machine Learning Repository. Available at https://doi.org/10.24432/C52P4X.

The code is available in the Supplemental File.

## Supplemental Information

Supplemental information for this article can be found online at http://dx.doi.org/10.7717/peerj-cs.2364#supplemental-information.

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
