# Peer review of "Kernel random forest with black hole optimization for heart diseases prediction using data fusion"

_PeerJ Computer Science, doi:10.7717/peerj-cs.2364_

## Round 0.1 · original submission · Major Revisions

Based on the reviewer comments, the manuscript must be revised.

Reviewer 1 ·

Basic reporting

The manuscript entitled “Kernel Random Forest with Black Hole Optimization for Heart Diseases Prediction Using Data Fusion” has been investigated in detail. The paper proposes a hybrid algorithm, KRF-BHO, for heart disease prediction by fusing multi-sensor signals from wearable devices and classifying medical data. However, the paper lacks clarity in problem statement and methodology. There are some points that need further clarification and improvement:
1) The introduction lacks clarity and fails to clearly articulate the problem statement and the significance of the proposed solution.
2) The transition between different sections of the paper is abrupt, making it difficult for the reader to follow the logical flow of ideas.

Experimental design

The methodology section does not provide sufficient detail about the proposed algorithm and its implementation, hindering reproducibility and understanding.

The paper lacks sufficient detail regarding the algorithm's design, implementation, and evaluation methodology.

There is a lack of justification for the choice of the proposed algorithm (KRF-BHO) over existing methods, and the theoretical underpinnings of the algorithm are not adequately explained.

Validity of the findings

The evaluation metrics used to assess the performance of the proposed algorithm are not clearly defined, and there is no comparison with existing state-of-the-art methods.

The experimental evaluation lacks rigor and detail, with limited information provided about the datasets used, experimental setup, and validation methodology.

There is a lack of discussion about potential limitations, assumptions, and biases in the experimental design and analysis.

The use of technical terminology is inconsistent, and there is a need for better clarity and precision in describing the proposed algorithm and its components.

“Result & Discussion” section should be edited in a more highlighting, argumentative way. The author should analysis the reason why the tested results is achieved.

The study named "Overcoming nonlinear dynamics in diabetic retinopathy classification: a robust AI-based model with chaotic swarm intelligence optimization and recurrent long short-term memory" can be used to explain of the proposed hybrid model processes if they wish.

This study may be proposed for publication if it is addressed in the specified problems.

Additional comments

The manuscript entitled “Kernel Random Forest with Black Hole Optimization for Heart Diseases Prediction Using Data Fusion” has been investigated in detail. The paper proposes a hybrid algorithm, KRF-BHO, for heart disease prediction by fusing multi-sensor signals from wearable devices and classifying medical data. However, the paper lacks clarity in problem statement and methodology. There are some points that need further clarification and improvement:
1) The introduction lacks clarity and fails to clearly articulate the problem statement and the significance of the proposed solution.
2) The transition between different sections of the paper is abrupt, making it difficult for the reader to follow the logical flow of ideas.
3) The methodology section does not provide sufficient detail about the proposed algorithm and its implementation, hindering reproducibility and understanding.
4) The paper lacks sufficient detail regarding the algorithm's design, implementation, and evaluation methodology.
5) There is a lack of justification for the choice of the proposed algorithm (KRF-BHO) over existing methods, and the theoretical underpinnings of the algorithm are not adequately explained.
6) The evaluation metrics used to assess the performance of the proposed algorithm are not clearly defined, and there is no comparison with existing state-of-the-art methods.
7) The experimental evaluation lacks rigor and detail, with limited information provided about the datasets used, experimental setup, and validation methodology.
8) There is a lack of discussion about potential limitations, assumptions, and biases in the experimental design and analysis.
9) The use of technical terminology is inconsistent, and there is a need for better clarity and precision in describing the proposed algorithm and its components.
10) “Result & Discussion” section should be edited in a more highlighting, argumentative way. The author should analysis the reason why the tested results is achieved.
11) The study named "Overcoming nonlinear dynamics in diabetic retinopathy classification: a robust AI-based model with chaotic swarm intelligence optimization and recurrent long short-term memory" can be used to explain of the proposed hybrid model processes if they wish.
This study may be proposed for publication if it is addressed in the specified problems.

Reviewer 3 ·

Basic reporting

The manuscript presents an approach, merging Kernel Random Forest with Black Hole Optimization to predict heart diseases using data from multiple sensors. The presentation of the algorithms is well-detailed, but it could benefit from a more narrative explanation that complements the technical description.

Experimental design

1. Provide a clear justification for the choice of the Black Hole Optimization Algorithm and how it compares or improves upon other optimization techniques.
2. Elaborate on the decision-making process for feature selection and its impact on the overall model performance.
3. Discuss any steps taken to prevent overfitting during model training, given the high accuracy reported.
4. Explain how the parameters for the Kernel Random Forest and Black Hole Optimization were chosen, as these can greatly impact the model's performance.
5. To improve the robustness of the study, consider adding external validation using datasets from different sources or geographical regions.
6. The paper would benefit from a more thorough comparison with current state-of-the-art methods beyond just a few mentioned algorithms. Detailed benchmarks against a broader set of existing techniques would enhance the paper's contribution to the field.
7. Future work could explore how the model performs in real-time settings and its responsiveness to data from continuous monitoring devices.
8. Many grammatical and typographical errors in the manuscript need to be revised for proofreading.

Validity of the findings

N/A

Additional comments

N/A

---

## Round 0.2 · accepted · Accept

Based on the reviewer comments, the manuscript can be accepted.

Reviewer 1 ·

Basic reporting

All my comments have been thoroughly addressed. It is acceptable in the present form.

Experimental design

All my comments have been thoroughly addressed. It is acceptable in the present form.

Validity of the findings

All my comments have been thoroughly addressed. It is acceptable in the present form.

Reviewer 3 ·

Basic reporting

N/A

Experimental design

N/A

Validity of the findings

N/A

Additional comments

The authors have thoroughly addressed all the comments, with some responses being well-justified. I have no further concerns.